# Fluoroquinolone-Induced Achilles Tendon Damage: Structural and Biochemical Insights into Collagen Type I Alterations

**DOI:** 10.3390/ijms262010028

**Published:** 2025-10-15

**Authors:** Magdalena J. Romanowska, Jakub P. Adamus, Sławomir Struzik, Leszek Paczek

**Affiliations:** 1Clinical Immunology Student Scientific Association, Department of Clinical Immunology, Medical University of Warsaw, Nowogrodzka 59, 02-006 Warsaw, Poland; jadamus.md@gmail.com; 2Department of Orthopedics and Traumatology, Medical University of Warsaw, Lindleya 4, 02-005 Warsaw, Poland; shoulder.slawomirstruzik@gmail.com; 3Department of Clinical Immunology, Medical University of Warsaw, Nowogrodzka 59, 02-006 Warsaw, Poland; leszek.paczek@wum.edu.pl; 4Institute of Biochemistry and Biophysics, Polish Academy of Sciences, Pawinskiego 5A, 02-106 Warsaw, Poland

**Keywords:** collagen type I, fluoroquinolones, fluoroquinolone-induced tendinopathy, Achilles tendinopathy

## Abstract

Fluoroquinolones, one of the most frequently used antibiotics, despite their wide spectrum of beneficial activity, are linked to serious adverse effects such as tendinopathies. Tendon injuries connected to the use of the group of drugs frequently affect the Achilles tendon—an anatomical structure, crucial to the proper mobility of lower limb, that is made of collagen fibers and extracellular matrix (ECM). Fluoroquinolones derive and decrease collagen and proteoglycans synthesis; they also disturb tendon regeneration by downregulating activity of metalloproteinases, enzymes essential for the proper collagen remodeling, especially after injuries. The exact way in which fluoroquinolones affect all these processes is not clearly known. However, some studies present that the chemical properties of fluorine such as electronegativity and ability to chelate di- and trivalent metal ions are one of the possible explanations for the problem. Our review summarizes various concepts of fluoroquinolones’ impact on the Achilles tendon structure, particularly collagen type I. What is more, it emphasizes the risk factors for more frequent Achilles tendon damage and presents the potential preventive strategies associated with the usage of the antioxidants.

## 1. Introduction

Collagen is a fibrous protein that plays a key role in providing structural support and maintaining the organization of the extracellular matrix (ECM). Its structure comprises a triple helix formed of α-chains [1]. The amino acids distinguishing collagen from other fibrous proteins are Glycine (33%), Proline (13%), and derived ones: 4-Hydroxyproline (9%) and 5-Hydroxylysine (0.6%) [2]. The amino acid sequences define the specific type of collagen. So far, 28 types have been identified [3]. Despite their differences, all collagen types share repeated regions containing tripeptides: Glycine (Gly)-Proline (Pro)-Y and Gly-X-Hydroxyproline (Hyp), where X and Y represent any amino acids [2].

The complex collagen synthesis pathway begins in fibroblasts, where the polypeptide precursors are synthesized. Following intercellular synthesis, procollagen is secreted into the ECM, where it is modified to form the triple helix. Collagen’s high mechanical resistance makes it a crucial structural protein of numerous tissues, such as bones and skin (type I), cartilage (types I, IX), blood vessels, skin, and muscles (type III). Its vital role underscores the severity of disorders linked to its synthesis and breakdown, for instance, the Ehlers–Danlos syndrome and osteogenesis imperfecta [1].

The most common type of collagen, type I, found in tendons, is critical in maintaining the structural integrity of the calcaneal (Achilles) tendon [4]. The Achilles tendon is formed by the gastrocnemius and soleus muscles. Furthermore, it is attached to calcaneal tuberosity. Interestingly, its anatomical localization is directly tied to the myth of Achilles, whose only vulnerable spot was his heel, likely near the tendon’s connection to the bone. The calcaneal tendon spans the knee and ankle joints, making it essential for numerous movements in various plains [5]. The most prevalent injuries to the Achilles tendon include (1) ruptures, often due to overexercise in athletes and amateurs alike, and (2) tendinopathies of various origins. Both of the conditions frequently require surgical intervention [6].

Tendinopathies can arise not only from excessive physical activity but also from the use of certain medications, most notably fluoroquinolones [7]. The mechanism of these antibiotics is to inhibit topoisomerase II, thereby limiting the transcription and replication processes in pathogens. They are effective against a broad spectrum of both Gram-positive and Gram-negative bacteria [8]. Although fluoroquinolones are widely used and [7] generally well-tolerated, they can occasionally lead to adverse effects, such as central and peripheral neuropathy, a greater risk of arrhythmia, and retinal detachment [9]. Moreover, these antibiotics have been reported to cause collagen degradation, highlighting the risk of tendon rupture and other connective tissue disorders [9].

## 2. Collagen Type I

Collagen type I consists of two α1-chains and one α2-chain [2] and is primarily composed of repeating sequences with Gly occupying every third position. The presence of Pro, another key amino acid, causes the single α-chains to be present in a left-handed polyproline-II-type helix [10]. The collagen also contains nonstandard hydroxylated proteins, e.g., hydroxyproline and hydroxylysine, which are a result of posttranslational modifications. Additionally, collagen may be classified as a glycoprotein due to its carbohydrate component [2], typically glucose or galactose [1], covalently attached to the derived amino acid through an O-glycosidic bond [2].

The whole collagen synthesis process begins in the nuclei of fibroblasts, where specific genes for pro-α1 and pro-α2 chains are transcribed into mRNA. The mRNA is subsequently transported to the rough endoplasmic reticulum (RER) and translated into the pre-procollagen chain [1]. The pre-procollagen consists of three major domains: (1) the α-chain, (2) the amino-terminal peptide, and the (3) carboxy-terminal peptide [10]. Posttranslational modifications include the removal of the N-terminal signal sequence (converting pre-procollagen into procollagen), hydroxylation of specific proline (Pro) and lysine (Lys) residues—the process requiring molecular oxygen, Fe^2+^ ions, and ascorbic acid—and glycosylation of selected hydroxylysine residues. Procollagen is composed of a central triple-helical region and non-helical N- and C-terminal extensions (propeptides). Disulfide bonds form at the C-terminal extension, ensuring the most accurate alignment for triple helix formation [1]. Following these modifications, procollagen is transported to the Golgi complex, packed into secretory vesicles, and released into the ECM [1,10]. In the ECM, specific metalloproteinases cleave the N- and C-propeptides from procollagen, yielding tropocollagen [11]. Tropocollagen monomers then are able to associate through covalent parallel crosslinking, forming fibrils that provide the necessary strength for further elongation [1,11] (Figure 1).

The copper-dependent enzyme lysyl oxidase [11] deaminates certain amino groups of Lys side chains to aldehydes (allysine and hydroxyallysine). Newly formed aldehyde can spontaneously undergo nucleophilic addition with Lys and hydroxylysine residues in adjacent collagen fibers creating covalent cross-linking [1,2], which enhances structural integrity.

## 3. The Achilles Tendon—Structure and Injuries

The calcaneal tendon, commonly known as the Achilles tendon, is among the most significant structures tendons of the lower limb. It is crucial for a full range of leg motion, particularly in knee flexion, foot plantar flexion, and hindfoot inversion [5]. Structurally, the tendon is composed mostly of tenoblasts, tenocytes, and ECM. Tenoblasts are immature tendon cells characterized by abundant cytoplasmic organelles and high metabolic activity. Over time, they elongate and differentiate into tenocytes, which present decreased metabolic activity [12]. Despite the reduction, tenocytes synthesize collagen type I, which accumulates for 70% to 80% of the tendon’s dry weight [13] as well as all ECM components [14]. The tendon’s low metabolic rate and well-developed anaerobic energy generation pathways enable it to withstand loads and maintain long-lasting tension while simultaneously minimizing the risk of ischemia and necrosis. However, these properties also contribute to slow post-injury healing [15].

The most prevalent Achilles tendon injuries include tendinitis, rupture, and tendinopathy [16]. Epidemiological studies indicate that the incidence of Achilles tendon ruptures has risen markedly over the past few decades [17]. Over 50% of tendon injuries occur during sports activities [18], with approximately 52% of runners having experienced Achilles tendinopathy at some point [19]. These injuries result from overload acting and repetitive microtrauma, exceeding the tendon’s physiological threshold [18], leading to sheath inflammation, structural degeneration, or a combination of both [15]. The most common types and location of the Achilles tendon injuries are presented in the Figure 2.

Degenerative tendinopathy is the most common histological finding in spontaneous tendon ruptures. Studies reveal that degenerative changes are present in 97% of spontaneous tendon ruptures compared to only 34% in control tendons [20]. The exact etiology of tendinopathy remains unclear, though numerous theories have been proposed, including hypoxia, the role of inflammatory mediators, fluoroquinolone exposure, and matrix metalloproteinase imbalance [15]. Tendinopathic tendons exhibit diffuse structural changes, such as increased tenocyte apoptosis, disorganization of collagen fibers with decreased collagen type I synthesis but increased type III collagen production, and ineffective neoangiogenesis [21]. Histological analysis shows greater degeneration in ruptured Achilles tendons compared to chronically painful tendons caused by overuse injuries [15].

Collagen type I, the primary component of the tendon matrix, demonstrates a high resistance to proteolytic cleavage [22]. However, collagenases, enzymes within the matrix metalloproteinase (MMP) family, are among the few capable of cleaving intact fibrillar collagen [23]. Researchers suggest that MMP-9 and MMP-13 participate exclusively in collagen degradation, whereas MMP-2, MMP-3, and MMP-14 are involved in both collagen degradation and collagen remodeling [24].

Tendon repair can be divided into three phases: inflammation, repair/proliferation, and remodeling [25]. During the inflammatory phase, erythrocytes, and inflammatory cells, predominantly neutrophils, migrate to the site of injury. Within the next 24 h, monocytes and macrophages begin to phagocytose necrotic material [15]. These cells are gradually replaced by fibroblasts, initiating the deposition of a type III collagen scar [25]. After six weeks, the remodeling phase begins, during which the healing tissue undergoes resizing and reshaping. Thus, the transition from a cellular to a fibrous structure takes place. This stage also involves intensified collagen type I synthesis. After 10 weeks, tenocyte metabolism and tendon vascularity decrease [15].

## 4. Fluoroquinolones and How They Affect Collagen

### 4.1. General Characteristics of Fluoroquinolones

Fluoroquinolones are among the most frequently prescribed antibiotics, mainly thanks to their exceptional pharmacokinetic (PK) and pharmacodynamic (PD) profiles, broad-spectrum antibacterial activity, relatively good tolerance [9], excellent tissue penetration, and, last but not least, their long half-life [26]. They belong to a class of drugs derived from quinolones, which were first discovered in 1962. Since their discovery, quinolones have evolved from being limited to treating Gram-negative urinary tract infections to becoming antibiotics with widespread activity, crucial in modern bacterial infection management [27]. Fluoroquinolones act by inhibiting DNA gyrase and topoisomerase IV, the enzymes that induce transient double-strand DNA (dsDNA) breaks [16], thereby directly interfering with proper bacterial dsDNA synthesis [28]. These drugs create a drug–enzyme–DNA complex, in which the type II topoisomerase is trapped with the covalently bound DNA, preventing the enzyme from cleaving, bonding, and rejoining DNA strands [16]. This process depends on the formation of a water–metal ion bridge, mediated by a Mg^2+^ ions, which connects the oxygen molecules in the amine group of the antibiotic with the hydroxyl residues in conserved serine or acidic residues in the enzyme [26,29].

The development of quinolones has followed two pathways, i.e., naphthyridones and fluoroquinolones, both characterized by a fluorine substitution at the 6-position [27]. The antimicrobial activity of each type of quinolone increases across their generation. Currently, five generations of fluoroquinolones and naphthyridones are recognized: I, IIa, IIb, IIIa, IIIb, and IIIc [27], or in another classification, 1, 2a, 2b, 3, 4, and 5 [30] (Figure 3).

### 4.2. Fluoroquinolone-Induced Tendinopathy—Epidemiology

Although fluoroquinolones presented good tolerance in randomized clinical trials, further epidemiological studies have linked them to a higher risk of serious adverse effects (AEs), including neurotoxicity and collagen degradation [9], especially in generation IIa (e.g., ciprofloxacin, ofloxacin, levofloxacin) [27]. The Achilles tendon is the structure most commonly affected by collagen malformations resulting from the use of fluoroquinolones. The tendons’ injuries have an overall relative risk of 3.2 cases per 1000 patients while currently using fluoroquinolones [31]. It is reported that during the first month of fluoroquinolone therapy, tendinopathy occurred in 85% of patients [32], but some patients may experience symptoms up to 6 months after discontinuation of antimicrobial treatment [33]. Ciprofloxacin has been associated with an elevated risk of tendinitis and tendinopathy, one of the earliest reported AEs [9], whereas levofloxacin is connected with tendon ruptures [28], most commonly affecting the Achilles tendon in 90% of cases [9].

In up to 50% of patients, symptoms of tendon injuries manifest bilaterally [28], typically appearing approximately 13 days after initiating quinolone therapy, and they may not subside for 12 months after treatment discontinuation [9,28]. This suggests an inability to restore tendon homeostasis [9].

### 4.3. Fluoroquinolone-Induced Tendinopathy—Possible Pathogenesis

Fluoroquinolone-associated tendinopathy involves collagen disruption and degeneration and a proteoglycan synthesis reduction, mainly by downregulation of MMP-degrading enzymes [9]. However, the exact mechanism by which fluorine affects protein molecules, particularly collagen, in fluoroquinolone-induced tendon injuries remains unclear [34] (Figure 4).

Tendinopathy associated with ofloxacin and levofloxacin treatment appears to be dose-dependent. Gradually increasing concentrations of levofloxacin lead to structural changes in the ECM, including reduced matrix production and elevated levels of MMPs. However, apoptosis occurs even at the lowest doses of the antibiotic [35].

#### 4.3.1. Chemical Characteristics of Fluorine

One of the hypotheses is that fluorine, with a dimensionless electronegativity value of 3.98 on the Pauling electronegativity scale, forms highly stable, polar covalent bonds due to its significant difference in electronegativity. Fluorine can also modulate the electron density of surrounding functional groups [36]. Most fluoroquinolone molecules exist as zwitterions, with an isoelectric point of 6.8–7.8, meaning their dominant forms at physiological pH are zwitterions [37]. Furthermore, fluoroquinolones are able to complex with orally administered divalent and trivalent metallic cations, such as the already-mentioned Mg^2+^, forming water-insoluble complexes. This interaction is pH-dependent [38]. The complexes with polyvalent cations can limit diffusion through the epithelial barrier, leading to subtherapeutic antibiotic concentrations [38].

#### 4.3.2. Disruption of Protein Synthesis

Fluorine in fluoroquinolones is believed to be responsible for ion chelation, especially Fe^2+^ in the context of collagen in the Achilles tendon. Fe^2+^ ions are essential in the activity of iron-dependent enzymes like prolyl 4-hydroxylase and lysyl hydroxylase, which are crucial for posttranslational modification of collagen. These enzymes mediate the hydroxylation of Pro and Lys residues necessary for collagen cross-linking and tensile strength of the fibers [39], offering another potential mechanism for fluoroquinolone-induced Achilles tendinopathy.

Moreover, fluorinated amino acids are not typically recognized by the endogenous protein synthesis apparatus. This indicates that the collagen molecules are modified post-synthesis rather than during their formation [34]. Fluorination of proteins primarily enhances stability by increasing hydrophobicity, altering folding rates, resistance to proteolysis, and mechanical stabilization [34].

The tight helical packing of collagen makes it challenging to accommodate fluorine-induced changes. However, substituting hydroxyproline with 4-fluoroproline has been shown to significantly enhance collagen’s conformational stability [36].

#### 4.3.3. Increased Expression of Metalloproteinases

Fluoroquinolones may also have an influence on MMPs, a family of proteolytic enzymes responsible for ECM network remodeling [4,9]. The activity of MMPs is regulated by tissue inhibitors of metalloproteinases (TIMPs) [40]. The balance between MMPs and TIMPs is crucial for ECM homeostasis [40,41]. Ciprofloxacin has been proven to increase the expression of MMP-3 [32] and MMP-2 [40] in human Achilles-tendon-derived cells, leading to reduced collagen synthesis through inhibition of tenocyte proliferation [32] as well as degradation of collagen type I. However, it does not appear to affect TIMP-1 or TIMP-2 expression [40]. Elevated expression of MMP-2 is associated with Achilles tendinopathy, while MMP-9 expression is upregulated in Achilles tendon rupture [40].

#### 4.3.4. Oxidative Stress

Another possible mechanism of disturbing tenocyte metabolism, collagen synthesis, and function is that fluoroquinolones induce oxidative stress in tendon tissue. Pefloxacin is one such example. It is both a member of the fluoroquinolones family and is considered to have that deteriorating impact, particularly when administered in large doses, essential for desirable drug effects [42].

The fluoroquinolone-induced collagen modifications are similar to alterations caused by ischemia–reperfusion, a model of oxidative injury in Achilles tendon tissue. Simultaneously administrated N-acetylcysteine, a well-known antioxidant [43], prevents the collagen from deleterious modifications. This observation suggests the reactive oxygen species (ROS) involvement in drug-induced tendinopathy [42]. Cells directly exposed to hydrogen-peroxide- or superoxide-generating agents may present an increased ratio of apoptosis [44]. In human tendon fibroblasts, this process mainly occurs in the course of caspase-3 activation and cytochrome c release from mitochondrial intermembrane space to cytosol. A lower concentration of ROS reveals degenerative activity in a time-dependent manner, whereas a higher concentration increases the rate of cell necrosis [45].

Tendon healing requires various processes, including migration of tenocytes to the healing area and synthesis of the ECM [46]. Ciprofloxacin, depending on administrated doses, inhibits tenocyte migration, primarily via the inhibition of focal adhesion kinase (FAK) phosphorylation [47].

Furuta et al. report that chronic pefloxacin usage increased the number of cells with round nuclei in a histological examination [48]. This observation is a characteristic feature of tendinopathy [49]. Noncellular changes indicating inflammation were observed. Present oxidative stress accompanies ROS, which were noticed to correlate with the increase in MMP number and activity [48]. A hierarchical link between ROS generation, the induction of matrix proteases, and further tendon degradation resulting in loss of tissue function were observed [50]. It is speculated that a higher ROS amount may be the cause of protease activation [50] and additionally may induce numerous pathways, including NF-κB and c-Jun N-terminal kinase (JNK) [48].

The expression of antioxidant enzymes genes such as *Gpx3*, *Gpx4*, *Sod2*, *Sod3*, and peroxiredoxin 3 was firmly reduced by pefloxacin administration. The main aim of the study was to examine genes responsible for the generation of GPX3 (glutathione peroxidase 3), a protein with selenocysteine in active site, which eliminates H_2_O_2_, hydroperoxides, and lipid hydroperoxides by supporting the conversion of glutathione to its oxidized form. It is important to highlight the fact that the promoter of the GPX3 gene contains a hypoxia-inducible factor-1 (HIF-1) binding site; therefore, it may be constantly activated in healthy tendon tissue because of its natural hypovascularity and low oxygen concentration. The study also showed that the gene expression of GPX3 was increased among antioxidant enzymes in human and rat tendons. Furthermore, this treatment paradoxically increased the expression of collagen-coding genes, mainly *Col1a1* and *Col3a1*, but it was considered as a compensation mechanism against progressive tendinopathic changes. These findings indicate that fluoroquinolones may be responsible for elevated oxidative stress in tissues during tendinopathy [48].

Fluoroquinolones are thought to decrease antioxidant enzyme activity and disrupt the cellular antioxidant status [51]. Peroxiredoxin 5 (PRDX5) is considered to be a protective agent against oxidative stress. It is synthesized in healthy tendons, but its amount increases in tendinopathy. This finding suggests that oxidative stress may underline the tendon disruption [52].

The supplementation of vitamin E and selenium, other examples of antioxidants [53], may have protective activity against fluoroquinolone-induced tendinopathy [32,53] by mitigating oxidative stress [54]. Moreover, they may reduce ECM degradation and apoptosis of tenocytes through decreased ROS accumulation [54]. Mișcă et al. stated that at the three-month follow-up, patients receiving such treatment experienced statistically significant lower pain scores compared to the controls. The scores were measured with the Visual Analogue Scale (VAS; *p* = 0.012), and better recovery was visualized in the Victorian Institute of Sport Assessment—Achilles scale (VISA-A; *p* = 0.034). The ultrasound imaging showed a lower tendon thickness and neovascularization, highlighting the advantageous effect of tendon remodeling [54].

The study results indicate the possible beneficial effect of antioxidant supplementation; therefore, more research should be focused on their impact on tendons during fluoroquinolone therapy. They may decrease the ratio of harmful changes in tissue, preventing tendinopathy or Achilles tendon rupture.

The summary of the possible mechanisms of Achilles tendon damage induced by the fluoroquinolones administration is presented in the Table 1.

## 5. The Cross-Section of Available Meta-Analyses

The dangerous adverse effects of fluoroquinolones have been examined in many meta-analyses. In particular, those related to tendon injuries have been on the main scope since 2008, when a black-boxed warning against an elevated risk of tendinitis and tendinopathy during fluoroquinolone therapy was released [55]. What followed was a great number of researchers reporting the harmful activity of fluoroquinolones in the tendon. Available meta-analyses mostly stratify patients based on age [35,55,56,57,58,59,60,61,62], gender [55,57,59,60,61], type of fluoroquinolone administered [35,55,56,57,58,60,61,62], dosage [35,55,58,60,61], time gap between exposure to antibiotics and the tendon damage [55,56,57,60,61,62], concomitant use of corticosteroids [35,56,57,58,61,62], and the occurrence of renal failure or hemodialysis in the past [35].

### 5.1. Influence of Age and Gender

When the analysis was stratified by age, there was a tendency of more frequent tendon damage amongst the elderly, mostly 60-year-olds [35,55,56,57,58,60,61]. Interestingly, Sangiorgio et al. 2024 advert to two studies about fluoroquinolones as a risk factor of tendon damage in childhood population and point that in this group, the most affected are children 3.3 ± 3.7 years old [60]. A few studies noted that fluoroquinolone-induced tendinopathy is more likely to be present among men [55,57,61], with a ratio of men to women of 1.16:1 [57] or 1.9:1 [61], whereas in other articles, females are more susceptible to develop tendinopathy [55,59]. What is more, two cohort studies revealed no association between particular gender and higher risk of tendon damage [57,59].

### 5.2. Stratification by a Fluoroquinolone Type

Although the toxicity of particular fluoroquinolones varies across the whole group and depends on the study, most researchers agree that levofloxacin [35,55,57] or ofloxacin [56,60,62], has the most harmful impact on Achilles tendon tissue. Still, some opinions are divided. There are studies concluding that a significantly increased rate of Achilles tendinopathy and rupture is associated with ofloxacin [56,60], while levofloxacin and ciprofloxacin had a lower impact on the tendon without statistically significant differences between each other [60]. On the contrary, other studies point to levofloxacin as the most frequent cause of tendon abnormalities in this antibiotic group [35,55,57] (Table 2). There are a few possible explanations why levofloxacin is included in large number of tendon rupture cases. These are older age of patients, concomitant use of corticosteroids, and a high rate of levofloxacin utilization in clinical practice [57].

### 5.3. Can the Dosage Alter the Risks?

Based on a number of studies, the average time of fluoroquinolone treatment is two weeks [55] with median of 9.5 days [60]. Khaliq Y. et al.’s meta-analysis revealed that most tendon damage occurs with a pefloxacin dosage of 800 mg per day with a median therapy duration of 9 days. In 25.5% of cases, ciprofloxacin was dosed in amounts ranging from 500 mg up to 2000 mg, primarily for 7 days. Further, there were cases with norfloxacin (11.2%) administered at 800 mg per day and levofloxacin (8.2%) and ofloxacin (6.1%) administered at 400 mg per day for 2–15 days [61]. These results stand in opposite to the reports stating that fluoroquinolones generally cause Achilles tendon impairment. The risk of any tendon damage is increased in patients receiving two or more fluoroquinolone prescriptions [58] or a cumulative dose of over 10,000 mg in the year before the index date [55,58] compared to those not using fluoroquinolones [58]. The risk of tendon rupture increases about 6% with each additional day of antibiotic administration [55].

### 5.4. Time Gap Between Exposure to Antibiotics and the Tendon Damage

Interestingly, the adverse effects of fluoroquinolones do not necessarily occur during or directly after antibiotic treatment. In some cases, the adverse effect, including tendinopathy, may happen after weeks [57] or months [55] of last dose of the medication. The mean gap between end of antimicrobial treatment and tendinopathy symptoms was noticed to be about 11 ± 5 days [60]. Most of the available meta-analyses, when focused on Achilles tendon damage, present it as a fluoroquinolone adverse effect occurring within 1 month after the last antibiotic dose [60,62]. There is a controversy in this topic, as some authors strongly recommend the wider period of time to be included in the studies [60], and others suggest that there is no statistical significance in the little available research, studying fluoroquinolone effects beyond a 1 month period [56].

### 5.5. Concomitant Use of Corticosteroids

Corticosteroids are a group of medications where concomitant application with fluoroquinolones may increase the risk of Achilles tendinopathy and rupture [58,62]. One meta-analysis revealed that these drugs were administered in 21.2% cases of tendon rupture, mostly accompanied with levofloxacin (27.1%) [57], whereas another reported that steroids were utilized in 32.7% cases of fluoroquinolone adverse effects and in 52.5% of cases of tendon rupture [35]. Drug regulatory agencies recommend avoiding this combination in order to decrease the potential of developing such damage [56].

### 5.6. Renal Failure or Hemodialysis in the Past

Approximately 90% renal elimination of levofloxacin as an unchanged drug may be the possible cause of its elevated toxicity in patients with renal dysfunction. Those people’s tendon tissues are exposed to a higher concentration of the fluoroquinolone for longer time. In a complex analysis of 28,907 fluoroquinolone-induced tendinitis and 7685 associated rupture cases, 1.4% and 2.2% of patients had renal failure or received dialysis. Despite the fact that the difference between patients with renal failure or dialyzed and those with normal renal activity was not statistically significant, a connection between impaired kidneys function and Achilles tendinitis was observed [35].

The summary of the clinical and epidemiological findings in the cited meta-analyses is presented in the Table 3.

## 6. Conclusions

Treatment with fluoroquinolones is linked to well-documented adverse effects, including markedly elevated risk of Achilles tendon injuries. While the general mechanisms underlying this damage are relatively well understood, the precise impact of fluorine on collagen type I molecules in the Achilles tendon remains insufficiently explored. As collagen type I is a key structural component of tendons, its degradation plays a critical role in the pathologies of fluoroquinolone-induced tendinopathy. Further research should prioritize investigating the interactions between fluorine and metal ions involved in collagen synthesis and remodeling. These processes may lead to decreased collagen density and resistance and therefore easier tissue damage. Moreover, the specific pathways by which metalloproteinases are regulated and influenced by oxidative stress in the presence of this chemical element should be better discovered. As long as inflammation, including an elevated activity ratio of the specific proteases, is prevalent, the discovery of particular preventive and therapeutic procedures may help to avoid tendon damage or further complications. Additionally, it is worth exploring if antioxidant supplementation may prevent the Achilles tendon from tendinopathy during fluoroquinolone treatment and directly after therapy. This kind of approach may be the solution for frequent tendon damage with no specific reason, e.g., mechanical injury, among patients with fluoroquinolone use in the past.

## Figures and Tables

**Figure 1 ijms-26-10028-f001:**
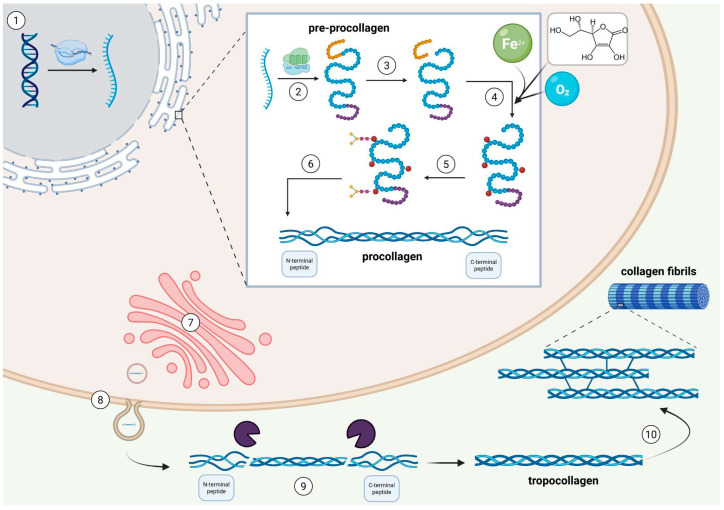
**The collagen synthesis pathway.** The complete collagen synthesis begins in the fibroblast nucleus with the transcription of specific genes for pro-α1 and pro-α2 collagen chains (1). Then, the mRNA is transported to the rough endoplasmic reticulum (RER) and translated into the pre-procollagen chain (2). The pre-procollagen consists of three major domains: the α-chain, the amino-terminal peptide, and the carboxy-terminal peptide. Subsequently, the posttranslational modifications occur, including the removal of the N-terminal signal sequence (3), hydroxylation of specific proline (Pro) and lysine (Lys) residues (4), and glycosylation of selected hydroxylysine residues (5). Created procollagen is composed of a central triple-helical region and non-helical N- and C-terminal propeptides (6). After these modifications, procollagen is transported to the Golgi complex, packed into secretory vesicles, and released into the ECM (7, 8). In the ECM, specific metalloproteinases cleave the N- and C-propeptides from procollagen, releasing tropocollagen (9). The cross-linking bonds between tropocollagen fibers are essential to form collagen fibrils and provide tensile strength (10). Figure created in BioRender. Romanowska, M. (2025) https://BioRender.com/vi29vha.

**Figure 2 ijms-26-10028-f002:**
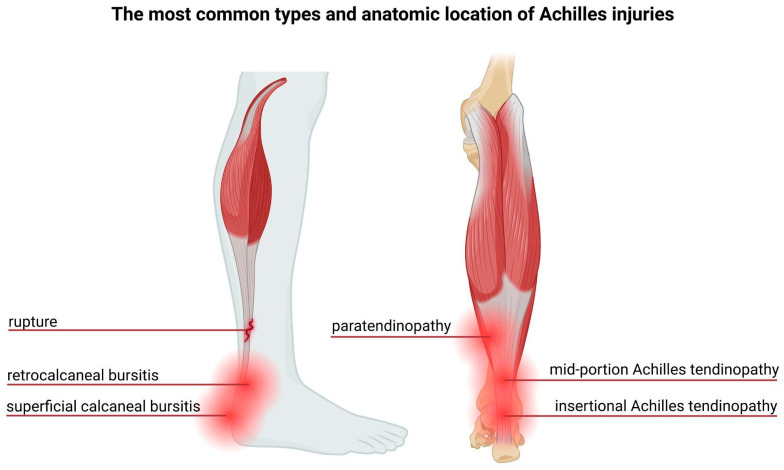
**The most common types and anatomic location of Achilles tendon Injuries.** There is no official terminology of Achilles tendinopathies, whereas clinicians use widely accepted ones, based on the anatomic location. This terminology includes mid-portion and insertional Achilles tendinopathy, paratendinopathy, and retrocalcaneal and superficial calcaneal bursitis. Except for the tendinopathies, one of the most common Achilles tendon injuries is a rupture. Figure created in BioRender. Romanowska, M. (2025) https://BioRender.com/gjb90ls.

**Figure 3 ijms-26-10028-f003:**
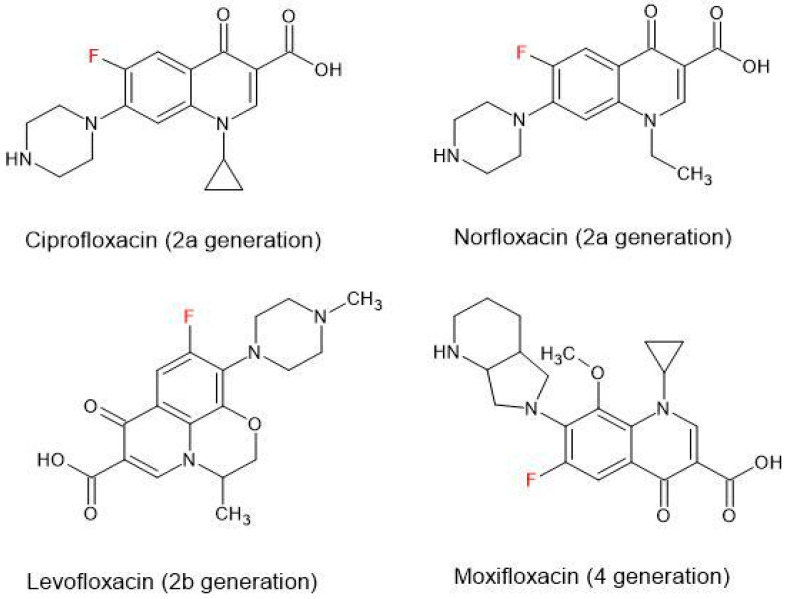
**The structure of selected fluoroquinolones.** 2a Generation: ciprofloxacin, norfloxacin; 2b generation: levofloxacin; 4 generation: moxifloxacin.

**Figure 4 ijms-26-10028-f004:**
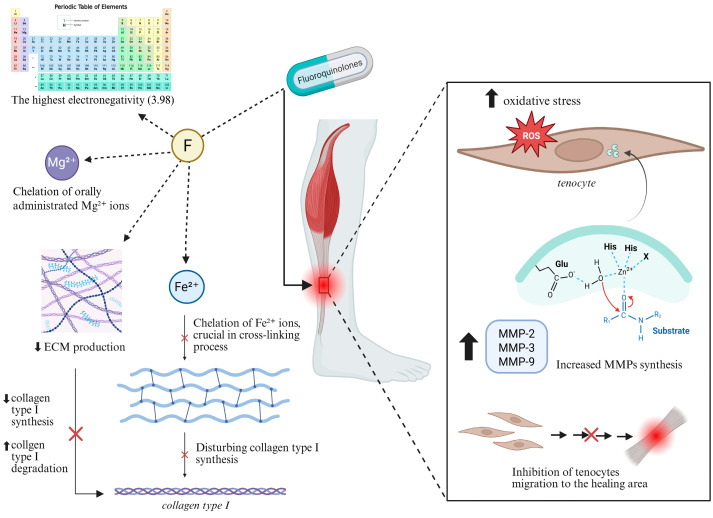
**Possible mechanisms of fluoroquinolone-induced tendinopathy.** Fluorine, as the key chemical element (the highest electronegativity in the Pauling scale, interactions with some divalent metal ions) in fluoroquinolones, may cause the Achilles tendon damage based on various pathways, including disruption of collagen synthesis, enzymatic interactions, decreasing ECM production, disturbing collagen type I synthesis and remodeling, increasing MMPs activity, slower tenocyte migration to the healing area, or a higher ratio of oxidative stress in the tendon. Figure created in BioRender. Romanowska, M. (2025) https://BioRender.com/l6p0x11.

**Table 1 ijms-26-10028-t001:** **Summary of the mechanisms of fluoroquinolone-induced damage.** Abbreviations: FQ—fluoroquinolone; ROS—reactive oxygen species; NO—nitrous oxide, MMP—matrix metalloproteases; ECM—extracellular matrix; MAPK—mitogen-activated protein kinase; oflox—ofloxacin; levo—levofloxacin; cipro—ciprofloxacin; nor—norfloxacin.

Mechanism	Takeaway Information
**Direct tenocyte toxicity**	Dose-dependent tenocyte apoptosis or necrosis is demonstrated in preclinical models.
**Oxidative stress (ROS/NO)**	FQs induce ROS/NO in tendon cells leading to matrix damage; a component of multifactorial injury.
**MMP and collagen synthesis**	Upregulated MMPs and reduced type-I collagen synthesis leading to ECM weakening.
**Mg^2+^** **chelation and integrin signalling**	Chelation impairs integrin/MAPK signalling and cell–ECM adhesion leading to impaired repair.
**Structure-toxicity link (C7)**	Higher tendon toxicity with C7 methyl-piperazinyl (oflox/levo) vs. piperazinyl (cipro/nor) in animal or toxicologic data.
**Mitochondrial or topoisomerase II** (speculative)	Possible host topoisomerase II or mitochondrial involvement; supportive but limited direct tendon data.

**Table 2 ijms-26-10028-t002:** **Fluoroquinolone-specific clinical significance.** Abbreviations: FQ—fluoroquinolone; ATR—Achilles tendon rupture; AT—Achilles tendonitis; levo—levofloxacin; cipro—ciprofloxacin; FAERS—FDA Adverse Effect Reporting System.

Fluoroquinolone	Takeaway Information
**Ofloxacin**	**Highest documented AT/ATR incidence** among common FQs: **≈1.40%** (95% CI, 0.88%–2.03%; SE 2.51); significantly greater than levo or cipro (≈0.17% each; *p* < 0.0001).
**Levofloxacin**	**Expected risk ≈ 0.17%** (similar to cipro in pooled estimates); most FAERS rupture reports and strongest disproportionality signal among FQs.
**Ciprofloxacin**	**Expected risk ≈ 0.17%**; ATR not significantly increased in some pooled analyses, but class risk still applies.
**Norfloxacin**	Meta-analytic stratification shows **increased ATR risk** for norfloxacin (agent-specific signal).
**Moxifloxacin/others**	Tendon events reported but **smaller pharmacovigilance signals** vs. levo; pooled “other molecules” group ≈ 0.31%.

**Table 3 ijms-26-10028-t003:** **Summary of clinical and epidemiological findings in selected meta-analyses.** Abbreviations: FQ—fluoroquinolone; ATR—Achilles tendon rupture; AT—Achilles tendonitis; OR—odds ratio; PY—person-years; CKD—chronic kidney disease.

Finding	Additional Information	References
**Incidence & Warning**	Tendinopathy is uncommon in general use (~0.14–0.4%); class carries boxed warning for tendinitis/rupture.	[62]
**Risk vs. Non-use (meta-analysis)**	FQ exposure increases risk: ATR OR ≈ 2.5; AT OR ≈ 4.0; any tendon disorder OR ≈ 2.0 (all significant).	[56]
**Predominant Site**	~90% of reported cases involve the Achilles; other tendons less frequent.	[61]
**Often Bilateral**	Bilateral involvement is common in Achilles cases (~40–50%).	[61]
**Onset Window**	Onset typically early: median ~8 days; 50% ≤6 days; range 2 h to 6 months; peak within first ~30 days.	[61,62]
**Progression to Rupture**	A large share of events progress to rupture (~40% of collated cases).	[61]
**Absolute Excess Risk**	Any tendon: **+3.73/10,000 PY**; Achilles: **+2.91/10,000 PY**; FQ and steroid vs steroid alone: **+21.2/10,000 PY**.	[58]
**High-risk: Age ≥ 60**	Older adults have materially higher odds of tendon injury/rupture on FQs.	[56,58,62]
**High-risk: Corticosteroids**	Concomitant systemic steroids markedly amplify risk (largest single clinical co-factor).	[56,58,62]
**High-risk: Renal Impairment & Transplant**	CKD/dialysis and solid-organ transplant recipients show higher rates; dose adjustment/avoidance advised where possible.	[59,61,62]
**Sex Signal**	Mixed: one large analysis found **higher odds in women** on FQs; early case series skewed older men.	[58,61]
**Drug-specific Risk Factor**	**Ofloxacin** use identified as a clinical risk factor for Achilles tendinopathy in cohort-level evidence.	[59]

## Data Availability

No new data were created or analyzed in this study. Data sharing is not applicable to this paper.

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
