# Peer review of "Fluoroquinolone-Induced Achilles Tendon Damage: Structural and Biochemical Insights into Collagen Type I Alterations"

_ijms, 2025, doi:10.3390/ijms262010028_

Round 1

Reviewer 1 Report

Comments and Suggestions for Authors

The manuscript entitled “Fluoroquinolone-Induced Achilles Tendon Damage: Structural and Biochemical Insights Into Collagen Type I Alterations” presents a timely and clinically relevant review on the adverse effects of fluoroquinolones, with a particular focus on collagen type I alterations in the Achilles tendon. The subject matter is highly relevant to the readership of IJMS because it integrates molecular mechanisms with clinical outcomes and addresses an important safety concern related to a widely prescribed group of antibiotics. The review is generally well-structured and cites a broad range of both classical and recent literature.

In its present form, the manuscript is primarily descriptive and would benefit from a more critical evaluation of the literature. For example, the discussion of drug-specific risks such as those associated with ciprofloxacin, levofloxacin, and ofloxacin could be expanded to highlight discrepancies in reported toxicity and the possible explanations for these differences. Similarly, the epidemiological and meta-analytical evidence is detailed but presented in a fragmented manner. A comparative synthesis, ideally with a table that summarizes patient risk factors such as age, gender, fluoroquinolone type, treatment duration, concomitant corticosteroid use, renal impairment, and time between drug exposure and tendon injury, would substantially improve the clarity and value of this section.

Another limitation is the limited number of figures. The manuscript currently contains only two figures, which is not sufficient for a comprehensive review article of this scope. Additional schematics would greatly improve readability and impact. I suggest including a schematic of collagen type I synthesis and cross-linking that highlights the role of Fe²⁺ and ascorbate, an illustration of Achilles tendon anatomy and typical sites of injury, and a timeline of tendon healing phases with overlays of how fluoroquinolones may interfere at each stage. Furthermore, a summary table or schematic outlining protective strategies such as antioxidant supplementation would strengthen the practical relevance of the review.

The language and style also require refinement. Several sentences are repetitive or awkwardly phrased, and careful editing would help improve clarity. The conclusion could be strengthened by more explicitly identifying key gaps in current knowledge, particularly the precise molecular role of fluorine in collagen disruption and remodeling, and by outlining future research directions.

In summary, this review addresses an important and clinically relevant subject and is within the scope of IJMS. However, significant revision is needed before it can be considered suitable for publication. The authors are encouraged to expand the visual content, provide a deeper and more critical synthesis of the literature, improve the language for readability, and place stronger emphasis on unresolved questions and future perspectives. These revisions would substantially enhance the overall quality, clarity, and impact of the manuscript.

Author Response

Dear Reviewer,

Thank you for all your comments to our manuscript. Please see the attachment, where we responded to them and provided the explanations.

Kind regards

Magdalena Romanowska

Reviewer 2 Report

Comments and Suggestions for Authors

I have carefully read the whole review article about fluoroquinolone-induced Achilles tendon damage and collagen type I alterations. The authors summarize how fluoroquinolones may reduce collagen and proteoglycan synthesis, disturb tendon regeneration by downregulating metalloproteinases, and possibly cause oxidative stress and metal ion chelation effects. The manuscript also reviews epidemiology data, possible mechanisms (fluorine electronegativity, MMP imbalance, oxidative stress), and even discusses antioxidants as potential protective strategies. I think this topic is important and relevant for both basic science and clinical readers. Below I provide my suggestions for improvement, mainly about novelty and clarity.

Major:

  1. The paper could better emphasize how this review differs from earlier systematic reviews and meta-analyses (e.g., the 2019 Br Med Bull review, or the 2024 EFORT systematic review cited). At this stage, the contribution appears more like a modest update rather than a major breakthrough. Highlighting clinical translation potential (e.g., how antioxidant supplementation or drug-design modifications could mitigate tendinopathy risk) would strengthen its originality
  2. The mechanisms (fluorine chemistry, oxidative stress, MMPs, etc.) are interesting, but sometimes you write them as if they are already proven, while many are still only possible explanations. It would be helpful to point out more clearly which parts are supported by strong evidence, and which remain hypothetical. For example, the section on oxidative stress (page 7–8) cites studies showing N-acetylcysteine or vitamin E/selenium supplementation reducing tendon damage, which gives stronger experimental support. In contrast, the part about fluorine substitution in collagen (4-fluoroproline stabilizing collagen helix, page 7) seems more speculative and not directly demonstrated in tendon tissue. Highlighting this difference will make your review more balanced and critical. Also, the idea of antioxidants as prevention is interesting, but the evidence is still limited. Better to acknowledge these limitations.
  3. The review would benefit from a stronger visual or comparative synthesis of the literature. At present, much of the paper is in long text form, where epidemiology, mechanisms, and meta-analysis results are described one by one. This makes it informative but not very engaging. Adding a summary table that compares key aspects would help. For example: different fluoroquinolones (ciprofloxacin, levofloxacin, ofloxacin) and their reported tendon risks, or the main mechanisms (oxidative stress, MMP imbalance, fluorine chemistry) with their evidence strength (strong / moderate / speculative).

Minor:

  1. The paper is mostly clear, but sometimes too detailed, especially in the introduction about collagen. This section could be shortened, focusing more on tendon biology and drug-related problems. Some sentences are quite long; using shorter sentences would improve readability. Figures are good, but the legends should explain more so that readers can understand them without reading the full text.
  2. The authors use many references, including recent ones, which is good. But some citations are basic textbooks — it would be stronger to cite more primary research articles. Also, when discussing meta-analyses, the paper mostly lists them one by one. It would add more value to synthesize and compare them, showing what findings are consistent and where they differ (e.g., whether levofloxacin or ofloxacin carries higher risk).
  3. In the abstract, please state more clearly what is new in this review. In the conclusion, instead of mainly repeating earlier content, be more forward-looking. For example, discuss what research questions are most important to answer next, or how these insights could guide clinical decision-making.

Author Response

Dear Reviewer,

Thank you for your comments. Please see the attachment with our response and explanations.

Kind regards

Magdalena Romanowska

Round 2

Reviewer 1 Report

Comments and Suggestions for Authors

The authors have fully addressed the previous major revision comments. The manuscript now presents a well-structured, scientifically robust, and clearly written review.

  • The mechanistic section has been substantially improved, covering fluorine–metal ion chelation, MMP/TIMP balance, and oxidative stress pathways with appropriate references.

  • Epidemiological and clinical details are more comprehensive, with clear data presentation in updated tables.

  • Figures, tables, and references are consistent and well-formatted.

  • The discussion and conclusion sections are coherent and meet IJMS standards.

Only a few minor typographical or language fixes remain, which can be handled by MDPI copy-editors prior to acceptance:

Minor Language Issues

  • Abstract (line 15): change “vide spectrum”“wide spectrum.”

  • Abstract (line 18): change “extra cellular matrix (ECM)”“extracellular matrix (ECM).”

  • Line 49: change “Achilles is formed by the tendons…”“The Achilles tendon is formed…”

  • Line 352: change “Sangiorgio et al.in 2024”“Sangiorgio et al., 2024.”

  • Throughout: add commas before “which” or “that” where needed for clarity.

Recommendation:
The manuscript is scientifically sound and suitable for publication in IJMS after minor editorial polishing.

Comments on the Quality of English Language

Minor Language Issues

  • Abstract (line 15): change “vide spectrum”“wide spectrum.”

  • Abstract (line 18): change “extra cellular matrix (ECM)”“extracellular matrix (ECM).”

  • Line 49: change “Achilles is formed by the tendons…”“The Achilles tendon is formed…”

  • Line 352: change “Sangiorgio et al.in 2024”“Sangiorgio et al., 2024.”

  • Throughout: add commas before “which” or “that” where needed for clarity.

Author Response

Dear Reviewer,

Thank you very much for your comments regarding English language fixes. We have made several improvements to enhance the quality and readability of the manuscript. 

Comment 1: 

  • Abstract (line 15): change “vide spectrum” → “wide spectrum.”

  • Abstract (line 18): change “extra cellular matrix (ECM)” → “extracellular matrix (ECM).”

  • Line 49: change “Achilles is formed by the tendons…” → “The Achilles tendon is formed…”

  • Line 352: change “Sangiorgio et al.in 2024” → “Sangiorgio et al., 2024.”

  • Throughout: add commas before “which” or “that” where needed for clarity.

Response 1: Every listed sentence have been corrected following your suggestions and the commas have been added in some places to improve the clarity of the manuscript.